# Research Progress of Nano Copper Lubricant Additives on Engineering Tribology

**Junde Guo** [1,2,*], **Yingxiang Zhao** [1], **Biao Sun** [1], **Puchao Wang** [1], **Zhijie Wang** [1] and **Hao Dong** [1,*]

1   School of Mechatronic Engineering, Xi'an Technological University, Xi'an 710021, China; zhaoyingxiang@st.xatu.edu.cn (Y.Z.); sunbiao@st.xatu.edu.cn (B.S.); jdgcxy-zyd@xatu.edu.cn (P.W.); wangzhijie@st.xatu.edu.cn (Z.W.)
2   National United Engineering Laboratory for Advanced Bearing Tribology, Henan University of Science and Technology, Luoyang 471000, China
*   Correspondence: gjd0119@xatu.edu.cn (J.G.); donghao@xatu.edu.cn (H.D.)

**Abstract:** Nanoparticles have as characteristics super sliding, extreme pressure, self-healing, etc., which can improve the friction reduction and anti-wear performance of sliding components, when used as lubricating oil additives. Nano-copper particles have a good synergistic effect with other antifriction agents, anti-wear agents, antioxidants and grease additives because of their low shear strength and grain boundary slip effect, showing a better anti-friction and anti-wear effect. However, nanoparticles are prone to conglomerate, and this causes a bottleneck in the application of dispersant for nano-copper in a lubricating oil system. The regulation of nanosized effect and surface properties has great engineering significance in compensating for the precision in manufacturing accuracy. This paper comprehensively reviews the tribological research progress of nano-copper as a lubricant additive, which provides a reference to the application of nano-copper particles as lubricating oil additives on engineering tribology.

**Keywords:** nano-additive; lubrication oil; friction and wear; lubrication mechanism

## 1. Introduction

With the enhancement of the environmental protection, the significance of green lubricating oil additives is growing, to meet the harsh restrictions on energy-saving and emission-reduction under the strict requirements of anti-friction and anti-wear [1–3]. Nano-scale additives applied to lubricating oil have great advantages in the field of precision manufacturing equipment because of the character of their surface–interface effect and high specific surface area [4]. In the process of operation of mechanical equipment, even if it is under the desired liquid lubrication condition, the interface of a friction pair can be in the boundary lubrication state under the influence of impact load, heavy load, low viscosity and repeated start-and-stop factors [5]. Although the traditional lubricating oil additives can improve the lubrication performance of tribo-pairs, their high reactive activation energy results in tribochemical corrosion wear, and their application is greatly limited in the friction parts of high-precision equipment [6]. Compared with traditional lubricating oil additives, nanoparticles have the unbeatable advantage of low film forming energy barrier, no film forming induction time, quick-forming of boundary protective film and micro-bearing effect on the friction surface–interface, further reducing equipment maintenance costs and improving the anti-friction and anti-wear performance of sliding components [7]. The research on nano-copper as lubricant additive has greatly progressed in the field of tribology, but nano-copper easily loses the characteristics of nanoparticles because the surface activity is relatively high, the primary cause being that a large number of unsaturated bonds exist on the surface of nanoparticles. In addition, the agglomeration of nano-copper, which is caused by VDW (Van der Waals' force), electrostatic action and hydrogen bonding between particles, increases the size of secondary forming particles and

loses the characteristics of nanoparticles. The above situation has a negative effect in the stability of nano-lubricating oil systems and easily causes oil circuit blockages in industrial applications. Therefore, it is important to prevent agglomeration of nano-copper particles, involving suspension time improving, dispersion stability and the tribological properties of nanoparticles in lubricating oil for nano-lubrication technology [8].

In this paper, the main research progress of nano-copper as lubricant additive is summarized, including dispersion stability, tribological performance, lubrication mechanism and theoretical simulation calculation. The friction performance, dispersion stability, theoretical simulation and main existing key issues related to nano-copper particles as lubricant additives are summarized. However, there are still some controversial conclusions in the research of nano-copper as a lubricating oil additive. For instance, the unity of dispersion and tribological properties, and it is still not widely used in industrial applications. This paper provides theoretical basis and technical support for the tribological performance research of nano-copper lubricant additives.

## 2. Characteristics of Nano-Copper Particles

The crystal lattice of nano-copper is a face-centered cube structure (FCC), and the diffraction angles of 43.36°, 50.481° and 74.16° are the diffractions of (111, 200) and (220) crystal planes of this face-centered cubic structure, respectively, as shown in Figure 1. It has small size particles and a quantum tunneling effect [9]. In addition, the nanoparticles also have the advantages of low melting point (high temperature resistance), low shear strength (easy to produce sliding effect on the friction surface) and a wide range of temperature application [10]. Under the action of friction, nano-copper not only has the advantages of micro-ball bearing, self-repair and deposited film effect [11]. The strength, ductility and anti-wear properties of nano-copper particles are better than the traditional large-sized copper particle as a lubricating oil additive, which is beneficial to the engineering application in the field of tribology [12].

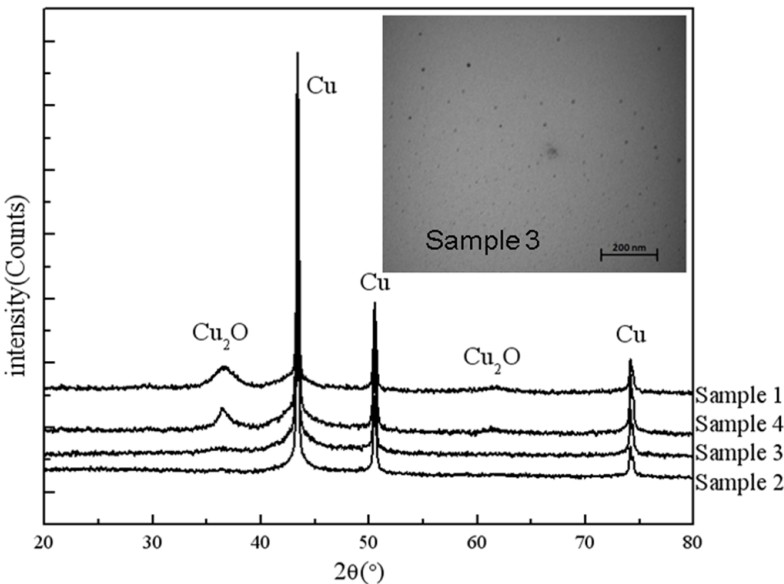

**Figure 1.** XRD and TEM pattern of nano-copper. Reprinted with permission from ref "Han, S.X. et al. Analysis and Optimization of Tribological Properties of Nano Copper Additive for Lubricating Grease. 2016". Copyright 2016, Xi'an Technological University.

## 3. Dispersion Stability of Nano-Copper Lubricant Additive

This section describes the current methods to improve the dispersion stability of nano-copper in lubricating oil and explains the dispersion mechanism of nano-copper particles in lubricating oil systems.

### 3.1. Surface Modification

The surface modification of nanoparticles is achieved by chemical and physical interactions between various surface additives and the surface of the particles and by the adsorption or reaction of various organic or inorganic chemical substances on the surface of the particles. Surface modification improves the surface state of particles, changes the surface properties of particles and produces new functions. After surface modification of nanoparticles, due to changes in surface properties, a series of properties such as adsorption, wetting, and dispersion will be changed to avoid the accumulation of nanoparticles. As shown in Figure 2, this method belongs to chemical cross-linking, and the modification layer of nanoparticles can cause spatial repulsion energy, prevent the interaction between particles and then improve the dispersion stability of nanoparticles in the base solution [13]. The commonly used modifiers are dispersants, inorganic electrolytes, surfactants, coupling agents and so on.

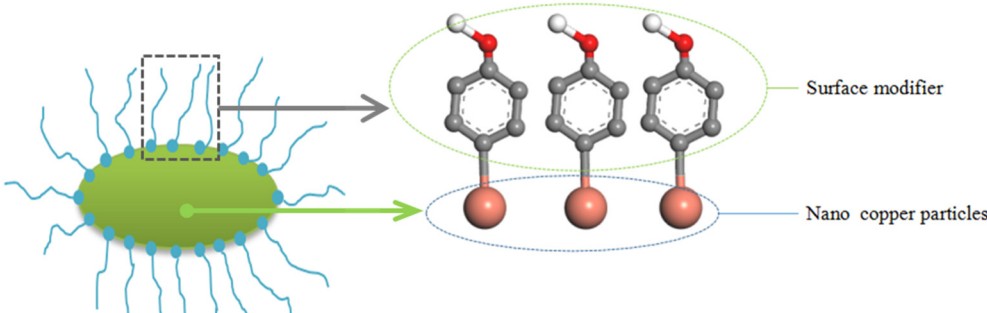

**Figure 2.** Diagram of nanoparticles for surface modification.

The stability of a lubricating oil system is closely relevant to the chemical adsorption force between the particles and the modifier. The existence of a large number of unsaturated and suspended bonds on the surface of the nano-copper leads to the low adsorption rate between the modifier and the particles, which affects the dispersion stability. Nanoparticles, such as multi walled carbon nanotubes, fullerenes, molybdenum disulfide and silicon dioxide, are used as additives of lubricating oil to form nano fluids, whose stability is affected by the base solution and the surface groups of nanoparticles suspended in the liquid [14]. Shaon et al. [15] prepared alkanols with long carbon chain, coated with nano-copper particles, and obtained good dispersion for nano-copper additive in base oil. Yang, H.J. et al. [16] prepared a nano-copper solution by chemical reduction and optimized the preparation process by changing the heating conditions, the amount of polyvinylpyrrolidone and cetyltrimethylammonium bromide, obtaining the preparation process parameters with good dispersion.

The dispersibility of nano-additive is mostly affected by the surface modification effect, which is not only related to the chemical modifier but also related to the structure of the nano-copper. Han et al. [17] obtained the freeze-dried nano-copper particles with good dispersion stability in the lubricating oil system by the surface modification method and freeze-drying process, using the modified nano-copper solution as the precursor, and the copper particle was subjected to rapid freeze-drying in liquid nitrogen. After optimizing the preparation process, the diffraction peak of cuprous oxide disappeared, indicating that the nano-copper particles with high purity were obtained. The freeze-dried samples have a high specific surface area, which is beneficial to the dispersion stability. Wang Z. and Wang Z. et al. [18] prepared nano-copper modified carbon nanotubes. After surface modification with dopamine, the friction coefficient of lubricating oil is the lowest when the nano-copper content is 0.2 wt%, which can reach 0.065. There is no obvious sedimentation phenomenon within 10 days, and the dispersion and tribological properties of the modified nano copper composite particles are the best. Zhang et al. [19] prepared nano-copper by DC arc method, modified and improved the nano copper by surfactant and achieved a

better dispersion effect by using the combination of oleic acid and dodecyl hydroxystearic acid and found that excessive ultrasonic treatment would lead to secondary agglomeration. The weak adsorption between hydrophilic group of modifier and nano-copper particle and the poor compatibility between hydrophilic group and base oil lead to the gradual deposition of nano-copper particle in lubricating oil with time, which leads to the unstable dispersion of nano-copper particles. Secondly, there is less research on base oil, because the base oil contains a large number of unknown additives, which cannot explain the essential reason for the stable dispersion of nano-copper particles additives. Moreover, the modified nano copper particle can only exist stably in a specific medium, which is not universal.

### 3.2. Polymer Coating Methods

The polymer coating method refers to the copolymerization reaction which used modifiers to prove the oil solubility of nanoparticles in the process of preparation and then improve the stability of lubricating oil. The common methods include surface modifier silicone modifier and methyl methacrylate [20]. Zhou et al. [21] coated Tween 80 and Span 80 by surface modification and blended with NO-460 lubricating oil to produce nano-copper composite lubricating oil with good dispersibility. Shi, S.C. et al. [22] studied the tribological behavior and energy dissipation properties of $MoS_2$ and nano-copper-reinforced biopolymer hydroxypropyl methylcellulose (HPMC) composites, and the results showed that the addition of $MoS_2$ and nano-copper particles improved the friction and wear properties of HPMC. Hosseini et al. [20] prepared copper nanoparticles with a diameter of about 75 nm by the galvanic substitution reaction of Pd ions with copper particles. Nano-copper particles were coated with cetyltrimethylammonium bromide (CTAB) in solution to reduce the surface tension of the particle and form a stable dispersion system. Carbon-coated nano-copper particles were prepared [23]. Peter et al. presented a method to embed copper particles into 6 different polymers, which is enlightening regarding composite particle preparation using polymer coating methods [24]. Ma et al. [25] prepared the nano-copper particles coated by PMMA and formed uniform Cu/PMMA composite microspheres, which significantly improved tribological performance of the base oil. Wang et al. [26] added surface-modified nano-copper particles to ZL101 heavy-duty gear oil to form a stable and uniform nano-copper composite heavy-duty gear oil system.

The microencapsulation method adds a uniform thin film of substances on the surface of the nanoparticle to form a core–shell structure by polymer coating, which is different from the surface modification method. Dong et al. [27] synthesized nano-copper microcapsules by chemical method and verified that the obtained nano-copper microcapsules could exhibit better polar pressure and tribological properties. Zhang et al. [28] obtained a longer lasting suspension stability than common nanocontaining copper in media using in situ polymerization to prepare nanocontaining copper microcapsules with a core–shell structure.

Yang et al. [29] prepared oil-soluble nano-copper particle with dioctylamine dithiocarbamate (DTC 8) as a modifier by two-phase extraction method, which did not cause discoloration or sedimentation when the non-polar solution was added and exhibited excellent dispersion stability. However, it was also found that the modifiers can lead to uneven size and weak controllability by chemical reduction and interfacial growth methods using modifiers of polyethylene glycol 2000 and Tween-85 [30,31].

### 3.3. Mechanical Dispersion Methods

The mechanical dispersion method belongs to physical dispersion in which the nanoparticles and dispersants are mixed and impelled by the rotation or vibration so that the adjacent nanoparticles can be physically rubbed and extruded, the nanoparticles and dispersants can be fully adsorbed and dispersed, and the dispersants can be successfully adsorbed to the surface of the nanoparticles [32,33]. The particle size of nano-copper particles can be refined and the contact area between modified materials and nano-copper can be increased by the mechanical dispersion method, but the particle oxidizes easily in the mixed or impelled process, while pulverization in liquid can protect nano-copper in

mechanical dispersion and reduce the degree of oxidation and further improve the purity of nano-copper products [34].

## 4. Tribological Development of Copper Composite Lubricants

Currently, the traditional lubricant additives hardly meet the requirements of extreme working conditions, including high loading and high strength [35,36], while nano-copper particles as lubricating oil additives have attracted much attention due to their unique physicochemical and polar pressure characteristics and self-repairing properties [37]. In addition, this has been validated under laboratory conditions as well as in industrial mechanical equipment. Jatti et al. [38] showed that the tribological properties of nano-copper oxide particles as additives in different kinds of lubricating oils show that the nano-copper oxide particles of base oil can reduce the friction coefficient in the mixed lubricant system by up to 50%. In the test bench test, adding 0.1 wt% nano-copper to API SJ 10W-30 gasoline engine oil, the friction coefficient can be reduced by about 29%, and the wear can be reduced by about 34%. In the real running environment of the car, the economy of nano-lubricating oil can be increased by 5% with the same mileage, and the average fuel consumption can be reduced by 1.44–3.09% [39]. Singh et al. [40] tested the friction and wear behavior of desert jujube oil with a nano-copper additive under different conditions by using a four-ball tester under laboratory conditions, and the results show that nano-copper particles can obviously improve the tribological property. The oleic acid-modified nano-copper was prepared by the liquid reduction method, and the friction characteristics and internal mechanism of nano-copper lubricants under current carrying conditions were studied through a modified four-ball friction and wear tester [41]. Gondolini et al. [42] synthesized nano-copper solubilizer by microemulsion and conventional precipitation method. When the prepared nano-compatibilizer was at 80 °C and the copper concentration was 0.01 vt%, the friction coefficient was reduced by 32.7%, and the anti-wear properties and extreme pressure properties were improved. The water-soluble nano-copper particle as pure water additive can improve the antifriction and anti-wear ability of pure water and effectively improve the friction performance of the water-based solution [43,44]. Wang et al. [45] prepared nano-copper by the liquid phase reduction method, and the corresponding friction coefficient and wear track under the load of 20 N with an average particle size of 128 nm were reduced by 25% and 54.4%, respectively, compared with the original LCKD-320 lubricating oil, and the anti-wear performance was remarkable. Sun et al. [46] prepared a reduced nano-copper solution with a particle size of about 139 nm, the friction and wear properties of nano-copper were evaluated by high-speed reciprocating tribometer. When the content of nano-copper in lubricating oil is 0.3%, the suspension time of nano-copper is longer. Under the pressure of 5 N, 10 N and 20 N, the friction coefficient is 67.07%, 15.79% and 43.40% lower than that of lubricating oil samples without nanoparticles. The extreme pressure performance and wear resistance have also been greatly improved. Zin et al. [47] used copper nanoparticles (NPs) as an internal combustion engine lubricant additive, and the results show that Cu NPs with a particle size of 130 nm have a good lubricating effect. Luo et al. [48] synthesized styrene-butylmethacrylate-3-methoxyacryloylpropyl trimethoxysilicon polymer coupling agent copolymer by free radical polymerization, and a macromolecular coupling agent was synthesized by free radical polymerization to bond with nano-copper. It can significantly improve the anti-wear and extreme pressure properties of the base oil at the concentration of 0.25 wt%.

An appropriate nanoparticle content can reduce the friction coefficient considerably, which is due to the nanoparticles affecting the lubrication state as shown in Stribeck curve describing the lubrication status in the hydrodynamic lubrication region. In the hybrid lubricating region, nanoparticles can form a boundary film on the friction surface, so that the Stribeck curve is left shifted, and the hybrid lubricating region is enlarged, which can meet the requirements of complex working conditions (Figure 3, $d_g$ represents the minimum oil film thickness and $R_a$ represents the surface roughness). The oil film between

the friction pairs separates the surface micro-convexities, and nanoparticles are not the main factor to affect lubrication state, but the addition of nanoparticles as lubricating oil additives can change the viscosity and thermal conductivity. This property is expected to obviously improve the lubrication state and, furthermore, reduce the maintenance/repair cost and the loss caused by shutdown and maintenance.

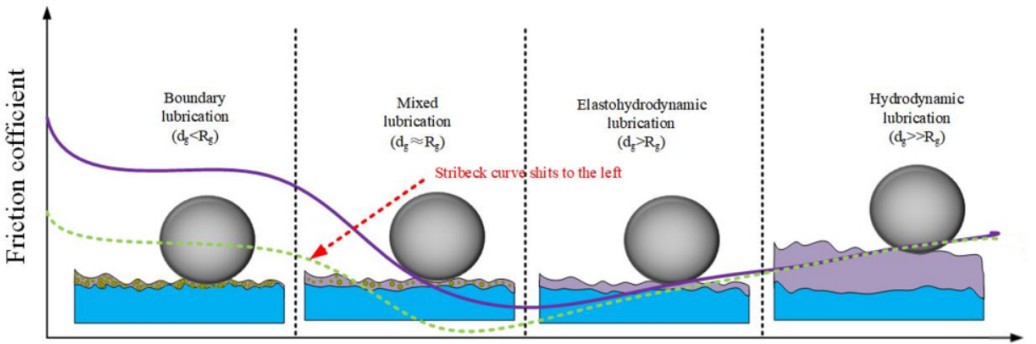

**Figure 3.** Effect of nano lubricant additive on Stribeck curve.

Proper addition of nano-copper can form a complete and uniform self-repairing film in friction and form boundary lubrication, as shown in Figure 4 (the base oil is paraffin oil). Excess lubricant additive will lead to uneven thickness of boundary film, causing abrasive wear and reduce the service life of lubricated parts. The excellent tribological performance of nano-copper particles as lubricating oil additive has been obtained from abundant theoretical supports [49]. Nano-copper particles can flow into the contact region during sliding process, which reduces the direct contact of friction pairs and thus reduces friction. Especially in the sliding friction condition of high load, the anti-friction effect is more prominent.

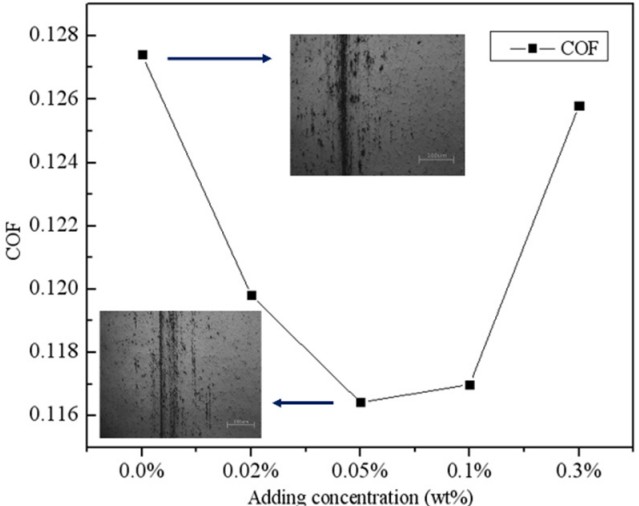

**Figure 4.** Tribological properties of copper nanoparticles at different concentrations. Reprinted with permission from ref. [17]. Copyright 2016, Xi'an Technological University.

In conclusion, the lubrication mechanisms of nano-copper are still not unified, and the main viewpoints include self-repairing mechanism, surface friction film-forming mechanism and micro-ball bearing effect. The theoretical simulation study is of great significance for the deepening of the anti-wear mechanism of nano-lubrication of lubricating oil additives in micro-scale.

## 5. Lubrication Mechanism of Nano-Copper as Additives

### 5.1. Forming Mechanism of Lubrication Film

After the nanoparticles arrive to the contact area in the sliding process, a deposition film or strengthening layer is formed by chemical and physical action, and the friction film can be formed on the sliding track, thus reducing the shear strength of the friction interface and the direct contact of the friction interface. In addition, it has a certain filling and self-repairing effect on the damaged parts of the surface (Figure 5). Liu Weimin modified copper nanoparticles (Cu-DDP, 6 nm) on the surface of dialkyldithiophosphoric acid (DDP) and tested their lubrication properties as lubricating oil additives. Due to the deposited copper nanoparticles having a small particle size, low melting point and good ductility, the atomic radii of Cu and Fe are very closed, and the covalent radii are the same, so a stable deposition film can be formed on the worn track since the low melting point and ductility under high temperature and high stress in the sliding process as shown in Figure 6. Good anti-wear, anti-friction and extreme pressure properties were obtained [50]. Based on the study of DLC solid–liquid composite lubrication system, Zhang et al. [51] pointed out that the high activity of nano-copper particles in NPCuDDP formed a lubricating film through tribological chemical reaction by comparing the tribological mechanism of nano-copper additive (NPCuDDP) and zinc dithiophosphate (ZDDP), which reduced the wear rate of all DLC coatings by 2–3 orders of magnitude compared with that of ZDDP or without additives. Yi D Z et al. [52] claimed that the improvement in the anti-wear ability and anti-fatigue performance of the steel–steel pair could be closely related to the special worn-surface repairing effect of the nano-Cu additive and the tribochemical reaction. Borda et al. [53] showed that the introduction of nano-additives can form an effective lubricating film on the sliding track and realize the transfer of contact stress, thus improving the anti-friction and anti-wear effect of lubricating oil.

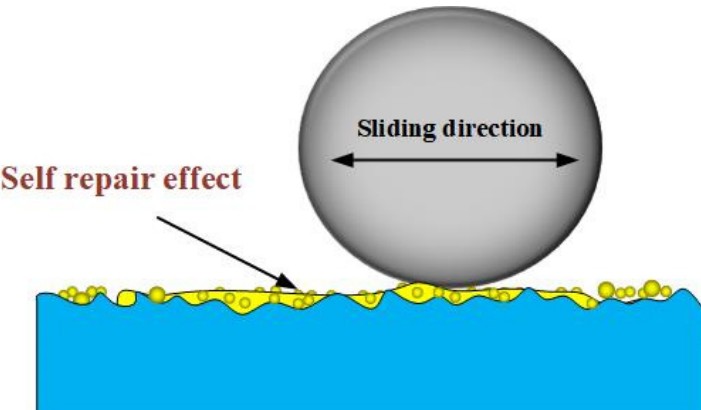

**Figure 5.** Schematic of lubrication enhancement of proposed lubrication mechanisms.

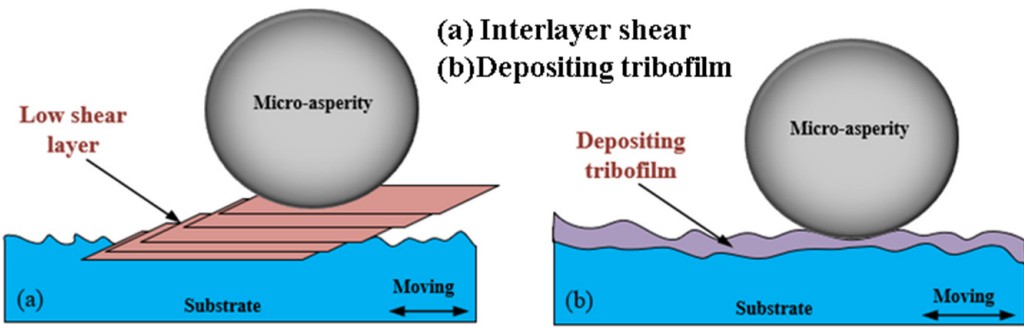

**Figure 6.** Mechanism of lubrication film forming.

However, there is also a different point of view that nano-copper softens to form a high viscosity fluid under the induction of friction heat in friction, which exerts the pressure-bearing lubrication effect on the friction interface to a certain extent [54]. Jiang et al. [55] prepared homopolymer polypropylene (PP-H) composite films with different nano-copper concentrations, presenting that the addition of nano-copper improves the thermal and mechanical properties of the composite film. Shi et al. [56] discussed the tribological mechanism of graphene/copper hybrid particles and hydroxypropyl methylcellulose composite solid lubricating coating, showing that the addition of graphene/Cu hybrid nanoparticles improves the lubrication performance.

### 5.2. Self-Repair Mechanism

Nano-copper particles with high activation characteristics are easy to adhere to the sliding surface or to bury in the micro-pit and micro-damage wear surface because of their large specific surface area and high surface energy, thus playing an effective protective role. Zhang et al. [57] think that the self-repairing effect is mainly accomplished by the coordination of physical, chemical and electrochemical action, further forming a better anti-wear effect. Yang et al. [58] observed a filling and self-repairing effect under the action of friction pair extrusion after adding 0.4% nano-copper to the lubricating oil, leading to the furrows and abrasions being significantly reduced on the sliding track; the deposition of copper nanoparticles plays a self-repairing role on the worn surface. Ye and Liu et al. [59,60] investigated the self-filling and self-repairing effects of the nano-copper precipitated from the surface defects of the coating since of copper nanoparticles and inferred that the friction heat will cause the nano-copper particles to absorb heat and produce an agglomeration effect, resulting in small aggregates of nanoparticles aggregating to form a lubricating boundary film during the sliding process [61]. Furthermore, the nano-copper particles can reduce the shear stress of the friction interface and weaken the adhesive and abrasive wear under high load, playing the role of lubrication protection on the friction surface [62].

The friction pair wear interface can constantly generate new metal surface during the sliding process, which is beneficial to forming the boundary film with self-healing effect. The surface metal activation caused by the tribo-chemical action is beneficial for the strengthening of self-healing effect in the application of nano-copper additives [63]. Shi et al. [64] verified that the friction chemistry is beneficial to the self-healing behavior of worn surfaces under the addition of modified nano-copper as self-healing materials. It has also been documented that nanoparticles can penetrate into the interior of the material lattice under frictional forces and play a self-healing role on the worn surface as shown in Figure 7, but this viewpoint lacks direct support and still needs to be refined and verified [65]. Wang et al. [66] conducted the tribological performance of nano-copper with self-healing additive in different lubricating oils, which showed that the repair film formed on the wear scar with the copper element. The self-healing effect of nano-copper is mainly related to the motion load and the content of nano-copper. If the load is too small, it is difficult to obtain the friction heat required for self-healing, but if the load is too large, the self-healing aggregates scarcely fall, resulting in the exacerbation of the lubrication state.

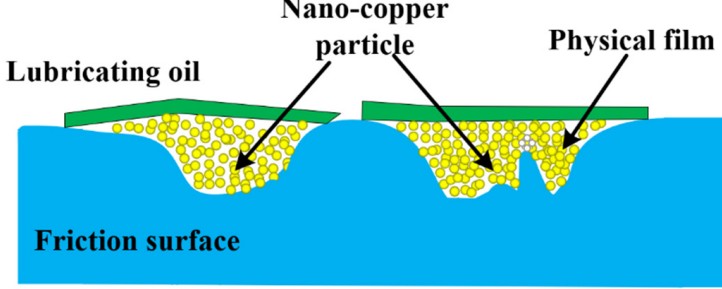

**Figure 7.** Self-healing mechanism.

### 5.3. Micro-Rolling Bearing Effect

Another opinion proposes that the thickness of the oil film under the boundary lubrication be increased after the nanoparticles enter the contact zone of sliding components and that the relative motion between the friction pairs be changed from the sliding friction state to the rolling friction state so as to avoid the direct contact between the friction pairs. Wang et al. [67] showed that the nano-copper particles can make the friction pair from the sliding state to the rolling state in the lubricating oil according to the experimental analysis, exhibiting good anti-wear and friction-reducing effects.

There are also certain differences in the effects of different scales of nano-copper on the performance of the lubricating fluid. In addition, excessive nano-copper can cause excessive wear because of the excessive size of agglomerate as shown in Figure 8. Because the spherical nano-copper is nearly spherical and the grains have dislocation distortion, the lattice will slip when a shear force arise, revealing that the nanoparticle resembles the ball bearing on the contact surface.

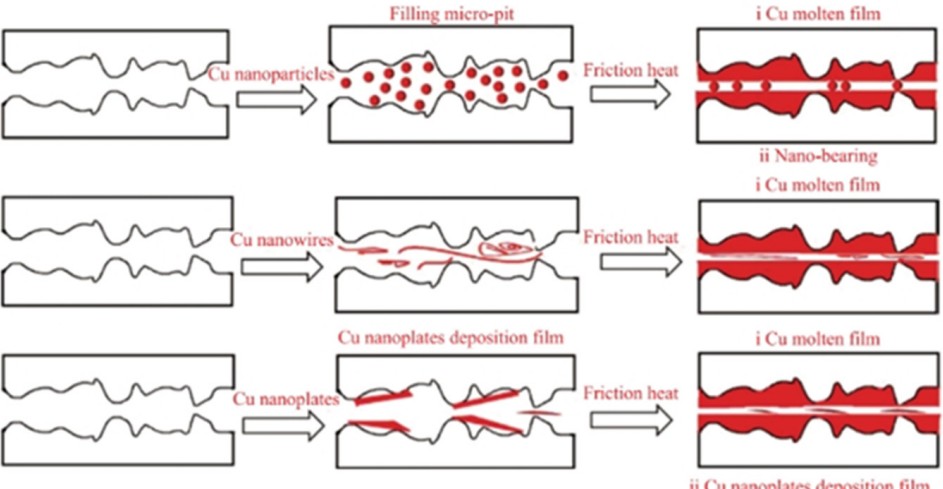

**Figure 8.** Lubrication mechanism of nano-copper lubricant additives with different sizes. Reprinted with permission from ref. [67]. Copyright 2020, China Petroleum Processing & Petrochemical Technology Press.

The spherical nanoparticles play a rolling role between the friction pairs, which can reduce the friction resistance, as shown in Figure 9. However, the application scope of the micro-ball bearing effect is effective under the low load condition, yet it can still maintain good rigidity under a heavy load and high temperature on the premise of well dispersed and spherical or similar to spherical particles. Although it is reported that the microsphere bearing effect can reduce friction for nano-copper, it still lacks direct experimental evidence for nano-copper as a soft metal [68].

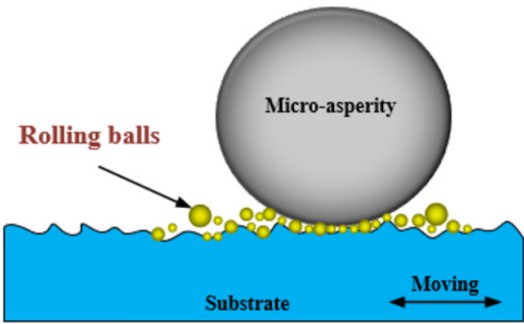

**Figure 9.** Schematic diagram of micro-ball bearing effect.

In addition, the load-carrying capacity of the nanoparticles can be affected by the shape or size. Chen et al. [69] considered that the key factors of particle lubrication are the ability of particles to enter the contact zone and whether the particles can stay in the contact zone. Wu et al. [70] indicated that flaky nanoparticles such as $MoS_2$ nano sheets are easily enter the contact zone of micro-convexity surface due to their small thickness, while spherical or linear three-dimensional particles are easily pushed away by micro-convexity because of their large size. Therefore, it is difficult to enter the sliding interface. The thickness of the boundary lubrication film is in the order of nanometers, which is much smaller than the comprehensive roughness of the common surface. If the diameter of the nanoparticles is larger than about 10 nm, it is more difficult to enter the gap (oil film thickness) between the tribo- pairs under the boundary lubrication. Therefore, the key to producing the lubrication effect is whether the nanoparticles can enter the contact zone [71].

At present, the anti-friction mechanism of "ball bearing" of nano-copper particles has been reported to some extent, but there is still a lack of direct experimental evidence. Because of its high viscosity, the effect of oil solubility of the lubricating oil can be ignored. Therefore, most soft metal particles can be added to grease as additives [72]. In the grease, the addition of nano-copper particles can improve the anti-wear performance and the service quality, which can bring fully into play the self-repairing of soft metals in the grease [73,74].

*5.4. Lubrication Theory and Simulation of Nanoparticle Lubricating Oil Additives*

At present, the lubrication theory, simulation methods and theoretical modeling of nanoparticles as lubricating oil additives are varied, mainly including molecular dynamics, CFD fluid simulation, mechanical modeling and ANSYS simulation. Particularly, molecular dynamics theory plays a guiding role in nanoparticles modelling, friction heat, nano-adsorption and boundary film formation. For the simulation of copper particles, fewer reports are related to tribology, and the main research focuses on molecular dynamics simulations of the aggregation of monocrystal and polycrystal nanoparticles [75], besides the simulation of common lubricating oil additives, such as nano-MoS2 [76].

Leng et al. [77] established the molecular dynamics model of nanoparticles by Lammps software and studied the effect of different sizes of the thermal conductivity of castor oil dispersion, which has a certain guiding effect on the mechanical properties and anti-friction mechanism of nano fluids. Through CFD simulation and physical modeling, Wu et al. [78,79] found that the dispersant polyisobutyleneamine succinimide (PIBS) plays a decisive role in the dispersion state and lubrication performance of the lubrication system. In the case of low PIBS percentage and no PIBS, nanoparticle aggregates can easily enter the friction contacted zone and form a uniformly distributed boundary lubrication film, resulting in reduced friction coefficient and wear. When a high percentage of PIBS is used to improve the dispersion state, there is a "flow around" effect of the contact point, resulting in poor lubrication performance. Zhou et al. [80] found that the introduction to additives forms an effective lubrication protective film on the friction surface; through ANSYS simulation and analysis of the worn surface, the local wear or micro-damaged surface can be self-repaired adaptively, thus improving the tribological performance.

## 6. Conclusions and Prospect

This paper reviews the research progress of anti-friction and anti-wear properties of nano-copper as a lubricating oil additive, in recent years. As a lubricating oil additive, nano-copper particles do not contain harmful substances such as sulfur and phosphorus, which is in line with the development direction of energy saving and environmental protection. The effectiveness and potential application of copper nanoparticles in improving the anti-friction and anti-wear properties of lubricating oil were presented, which has a good effect on the improvement of lubricating oil dispersion, stability and tribological properties, and is expected that copper nanoparticles become a new generation of industrial lubricating oil additives with potential.

In this paper, the factors affecting the tribological properties of nano-copper particles in lubricating oil system and the methods to enhance the dispersion stability and the lubrication mechanism of nano-lubricants are summarized. The agglomeration of nanoparticles is the main problem facing the long-term stability of lubrication systems, although a large number of reports claim that the dispersibility has been improved by surface modification, polymer coating and mechanical dispersion and so on. Nano-copper particles have better dispersion stability after modification or freeze-drying treatment, and most studies aim at the overall dispersion of lubricating oil with nano-copper particle additives. There is no unified conclusion on the nature of particle stability and lubrication mechanisms in different media. On the basis of maintaining good dispersion stability, it is still the main challenge to achieve the unity of dispersion stability and tribological properties for copper nanoparticles as lubricating oil additives. With the dispersion of nano-scale additives, it is difficult to form an acknowledged system, and the effectiveness of lubrication is uncertain under different working conditions; the understanding of the industrial circle of the lubrication mechanism of the nanoparticles is still in the speculative stage, and there is a lack of unified understanding and direct experimental evidence. It limits the popularization and application of nanoparticles as lubricating oil additives in industry.

As a soft metal, there has been some experimental evidence for deposited film formation and self-repair effect of nano-copper. In addition, the theoretical analyses of nanoparticle lubricating oil additives lack corresponding simulation and theoretical simulation methods for the synergistic lubrication performance of nanoparticles and common lubricating oil additives such as ZDDP. The lubrication condition of nano-copper oil additives mainly depends on the experimental means, and the suitable working conditions and evaluation methods of copper nanoparticles as lubricating oil additives have an important reference value for tribological performance optimization.

**Author Contributions:** J.G. conceived and designed this work; Y.Z., P.W. and B.S. prepared the original draft; Z.W. and H.D. provided the revision work. All authors have read and agreed to the published version of the manuscript.

**Funding:** This work was supported by the Project National United Engineering Laboratory for Advanced Bearing Tribology of Henan University of Science and Technology (202106); Science and Technology Project of Beilin District (GX2140); Special Research Project in Shaanxi Province Department of Education; Science and Technology on Diesel Engine Turbocharging Laboratory (6142212190104); Science and Technology Project of Weiyang District (202111); Science and Technology Project of Xi'an (21XJZZ0027).

**Institutional Review Board Statement:** Not applicable.

**Informed Consent Statement:** Not applicable.

**Data Availability Statement:** No data, models, or code were generated or used during the study.

**Conflicts of Interest:** The authors declare no conflict of interest.

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
