# Peer review of "Research Progress of Nano Copper Lubricant Additives on Engineering Tribology"

_metals, doi:10.3390/met11122006_

Round 1
Reviewer 1 Report
In the paper, “Research progress of nano copper lubricant additives on engineering tribology” authors review the tribological research progress of nano-copper as lubricant additive.
Although the manuscript is about an interesting subject and it could be considered as a reference, contains several aspects that must be improved in order to take into account this work for publication.
-References must be reviewed, because they are not in the same format. It is sometimes possible to read the given name of the authors but most times not. The use of capital letters shoud be also reviewed.
-English spelling must be check. There are several mistakes that prevent understanding the text: lack of verbs, capital letters, punctuation marks, repeated words...
-Could be possible to explain how it is possible to have “low melting point and high temperature resistance”? Line 63
-Please, rewrite lines 65-66 and 67.
- Definition of surface modification (line 80) is really poor, please, change it.
-I do not understand the difference between the two images in Figure 2.
- Why is polyvinylspyrrolidone and cetyltrimethylammonium bromide use to prepare copper nanoparticles? (line 94).
-Reference to Fig 1 (line 108) does not correspond.
-Author of reference [18] is not Song Haojie (line 108), according to the reference list authors are Wang et al.
-Could to explain what are Tween 80 and Span 80 (line 132)
-In lines134-138 there are no reference in the text to polymer coating methods.
-There are some references in the reference list that not appear in the text, as [24], [25], [55],[56],..
-I do not understand “15%Murray 45%” in line 186.
-“Nano-copper particles can obviously improve the friction coefficient” (line 193), in my view, it is not an obvious affirmation.
-I do not understand “volt” in line 201.
-In Figure 4 and Figure 7 please use English words.
-What lubricant is used in Figure 5.
-Please, include the reference of Liu Weimin (line 262).
-There is not Figure 11 (line 333 and line 346).
- I would like to know if there are any previous works about nano-copper particles simulation.
Author Response
Dear Reviewer:
Thank you for your kind comments concerning our manuscript entitled “Research progress of nano copper lubricant additives on engineering tribology” (Manuscript ID: metals-1450146). These comments are valuable and helpful for revising and improving our manuscript, as well as the important guiding significance to our researches. And we made cautious revision accordingly. Based on these comments, we have answered the questions in detail one by one. If you have any other questions about this paper, I would quite appreciate it if you could let us know in the earliest possible time.
We are looking forward to hearing from you. Any further information and suggestions are greatly appreciated.
Junde Guo
2021.11.06

Reviewer 2 Report
1) This review paper described the research on dispersing copper nanoparticles, which are soft metals, in lubricating oil, which is effective for improving friction and wear characteristics and preventing deterioration of lubricating oil.
In this paper, conventional methods and issues are systematically summarized. Specifically, they are the control methods, the effects on tribological properties, and issues of nanoparticle dispersion from various view points such as chemistry on dispersion and aggregation of nanoparticles, contact engineering using Stribeck curve, and simulation.
In addition, most of the cited papers are relatively new.
Based on the mentioned above, I’ve judged that this paper is useful in engineering as a review paper.
2) Since some Chinese characters are used in Fig. 4 and Fig. 7, they should all be unified in English.
Author Response

(The authors gave the same response as above.)

Reviewer 3 Report
Abstract should be completely rewritten. Background is too long. Abstract should consists background, review methods, findings and main conclusion(s). Describe briefly the main review methods applied. Please summarize the article's main findings and indicate the main conclusions or interpretations. https://www.mdpi.com/journal/metals/instructions
Methods of review have not specified. Review papers must provide concise and precise updates on the latest progress made in a given area of research. Systematic reviews should follow the PRISMA guidelines ( https://www.mdpi.com/journal/metals/instructions )
In the "Introduction" section the authors provided background information on lubricant additives. However, the scientific contribution/novelty is very thin and weak. The scientific novelty of the review work should be stressed out in this section.
Style of citing references does not comply with international standards and requirements of publisher. For example paper [15] should be referenced as "Shaon et al.", not "Shaon". Paper [16] should be referenced as "Yang et al.", not "Yang Haijun". etc. The whole manuscript needs to be checked. I encourage authors to view published articles in "Metals".
Caption of the figure 3 is inappropriate. This figure should be divided into (a), (b), ... and the caption should be modified accordingly.
line 185: "of 15% Murray 45% [38]" ???
line 187: "SJ" Each abbreviation must be define the first time they appear in the main text.
"Jiang Zichao [41] oleic acid modified nano-copper prepared by [...]" ???
"Luo [48] Styrene-butyl methacrylate-3-methoxyacryloylpropyl trimethoxysilicon polymer coupling agent copolymer was prepared [...] ???
THE LANGUAGE OF THE MANUSCRIPT NEEDS POLISHING THROUGHOUT.
Figure 4: It is not known what the parameters dg, Ra in this figure mean. Each parameter used in the manuscript must be clearly defined. Replace the Chinese characters in this drawing with English.
Figure 7: Replace the Chinese characters in this drawing with English.
Figure8: Caption of this figure is inappropriate. This figure should be divided into (a), (b), ... and the caption should be modified accordingly.
line 262: "Fig. 6-2" ?
line 333: "Fig. 6-3" ?
Figure 10 should be referenced in the main text before figure 11.
There are two figures named "Figure 6" in the manuscript.
List of references is not formatted according to "Instructions for Authors".
About 60-70% articles cited in the List of references are written by the authors from China.
I strongly believe that review has not been conducted properly. There are many research groups in the world dealing with similar topics. So, the article should take into account the main publications from other research centres. Otherwise, title of manuscript should be changed as "Research progress of nano copper lubricant additives on engineering tribology mainly in China".
Author Response

(The authors gave the same response as above.)

Round 2
Reviewer 1 Report
Authors of “Research Progress of Nano Copper Lubricant Additives on Engineering Tribology” have considered the suggestions provided and they have also answered kindly to the comments and doubts.
I have a few comments to improve the quality of the paper:
- English spelling must be reviewed (e.g., in line 53, “conclusionsins”, line 129 and line 164 “is refers”, “is belong”)
- I would propose to use “Nano-copper particles” in plural on the whole text. It is difficult to consider/study only one particle.
- All references in the text must be reviewed because it is necessary, not only mention the first author, but also include “et al” when there are more than two authors.
- The paragraph between lines 113-116 should not be in italics.
- [20](line 138), Ref [27] (line 151), Ref. [47] line 211, Ref [53] (line 273), Ref [66] line 313, Ref. [67] line 326, Ref [60] line 296 do not correspond with the reference list.
- Line 184, there are two similar references.
- Ref [38] and Ref [52] are the same.
- Please, add the type of lubricant base oil used in test represented in Figure 4.
Author Response
Dear reviewer,
Authors of “Research Progress of Nano Copper Lubricant Additives on Engineering Tribology” have considered the provided suggestions and we have answered carefully to the comments and doubts one by one. The 1st version has some problems on the modified trace display, we have corrected this and present a suitable version.
Best regards,
Junde Guo
- English spelling must be reviewed (e.g., in line 53, “conclusionsins”, line 129 and line 164 “is refers”, “is belong”)
Answer: Thank you for your comment. We have made changes in the corresponding position of the article
- I would propose to use “Nano-copper particles” in plural on the whole text. It is difficult to consider/study only one particle.
Answer: We have made 10 modifications to the original text according to your suggestion, about the plural form of "nano copper particles"
- All references in the text must be reviewed because it is necessary, not only mention the first author, but also include “et al” when there are more than two authors.
Answer: Thank you for your comment. We carefully reviewed all references in the text and corrected 31 first authors and 36 missing "etc".
- The paragraph between lines 113-116 should not be in italics.
Answer: We are sorry for this mistake, we have corrected this paragraph.
- [20](line 138), Ref [27] (line 151), Ref. [47] line 211, Ref [53] (line 273), Ref [66] line 313, Ref. [67] line 326, Ref [60] line 296 do not correspond with the reference list.
Answer: We have revised these questions to correspond to the list of references
- Line 184, there are two similar references.
Answer: We deleted redundant references [38]
- Ref [38] and Ref [52] are the same.
Answer: We changed Ref [52] to Ref [38] and adjusted the corresponding contents.
- Please, add the type of lubricant base oil used in test represented in Figure 4.
Answer: We added the base oil type, i.e. paraffin oil, in the text corresponding to figure 4.

Reviewer 3 Report
First of all, the authors did not prepare a revised version of manuscript according to the requirement of publisher: „(I) Any revisions to the manuscript should be marked up using the “Track Changes” function if you are using MS Word/LaTeX, such that any changes can be easily viewed by the editors and reviewers.”
The reason is that they included only minor changes in the manuscript.
- My comment no. 1:
"Abstract should be completely rewritten. Background is too long. Abstract should consists background, review methods, findings and main conclusion(s). Describe briefly the main review methods applied. Please summarize the article's main findings and indicate the main conclusions or interpretations. https://www.mdpi.com/journal/metals/instructions"
has not been incorporated in the manuscript.
The abstract still contains commonly known information that does not require a literature search. Conclusions should be drawn based on the literature review.
- On my comment no 2:
"Methods of review have not specified. Review papers must provide concise and precise updates on the latest progress made in a given area of research. Systematic reviews should follow the PRISMA guidelines (https://www.mdpi.com/journal/metals/instructions )"
the authors responded: "We have simplified and revised the whole paper accordingly, the modified content has been marked in DIFFERENY COLOR."
The authors ADDED ONLY ONE SENTENCE IN THE WHOLE MANUSCRIPT (!), marked in green in page 5 (lines 192-195).
This sentence is not completely related to my comment.
- On my comment
"In the "Introduction" section the authors provided background information on lubricant additives. However, the scientific contribution/novelty is very thin and weak. The scientific novelty of the review work should be stressed out in this section."
the authors answered: "In this section, we have made MAJOR revision to summarize the current research status and shortcomings, the value and significance of this paper are pointed out in this section."
It is very confusing answer because the authors linguistically changed only three first sentences in the „Introduction” section.
- My comment:
"Style of citing references does not comply with international standards and requirements of publisher. For example paper [15] should be referenced as "Shaon et al.", not "Shaon". Paper [16] should be referenced as "Yang et al.", not "Yang Haijun". etc. The whole manuscript needs to be checked. I encourage authors to view published articles in "Metals".
has not been incorporated in the manuscript. There are still the same problems.
- On my comment:
“List of references is not formatted according to "Instructions for Authors". About 60-70% articles cited in the List of references are written by the authors from China. I strongly believe that review has not been conducted properly. There are many research groups in the world dealing with similar topics. So, the article should take into account the main publications from other research centres. Otherwise, title of manuscript should be changed as "Research progress of nano copper lubricant additives on engineering tribology mainly in China".
the authors answered:
Thank you for your comments. We have replaced reference 47 and revised the content of its corresponding article. As shown in Figures a, b, c and d below, we made a statistic on Web of Science with nanoparticles lubrication as topic and Copper as the title and got 164 search results. Then refine the search results with PEOPLES R CHINA as the condition to get 81 results, which is 49.39%. Obviously, Chinese scholars have done a lot of research on copper nanoparticles lubrication.
I completely disagree with the conclusion that " Obviously, Chinese scholars have done a lot of research on copper nanoparticles lubrication."
The authors limited the scope of the review only to the Web of Science database. This has not been clearly pointed out in the manuscript. Another mistake is database search using topic "Copper nanoparticles lubrication" and "Cooper" in Title. The use of “copper” in both topic AND title significantly reduces the number of articles found. I conducted a similar analysis in Sciencedirect database and I found 3,435 results. LIMITING THE SCOPE OF THE SEARCH TO THE WEB OF SCIENCE DATABASE IS NOT PROBLEM FOR ME. However, considering the above, I disagree with the authors' statement in abstract that "This paper >>COMPREHENSIVELY<< reviews the tribological research progress of nano-copper.” Manuscript should be appropriately corrected for the scope of the review.
Finally, the authors added new figures without any justification of such a procedure.
========================
Any revisions to the manuscript after 1st and next review round should be marked up using the “Track Changes” function if you are using MS Word/LaTeX, such that any changes can be easily viewed by the editors and reviewers.
========================
Author Response
Thank you for your helpful comments. Please see the pdf file below.
